Identification and validation of apoptosis-related genes in acute myocardial infarction based on integrated bioinformatics methods

Zhu Haoyan 1 2
Li Mengyao 3 4
Wu Jiahe 1 2
Yan Liqiu 3 4
Xiong Wei 1 2
Hu Xiaorong 1 2
Lu Zhibing 1 2
Li Chenze 1 2 lichenze@126.com
Cai Huanhuan 1 2 caihuanhuan@whu.edu.cn
1 Department of Cardiology, Zhongnan Hospital of Wuhan University , Wuhan , China
2 Institute of Myocardial Injury and Repair, Wuhan University , Wuhan , China
3 Department of Cardiology & Dongguan Cardiovascular Research Institute, Dongguan Songshan Lake Central Hospital, Guangdong Medical University , Dongguan , China
4 Department of Cardiology, Cangzhou Central Hospital, Hebei Medical University , Cangzhou , China
Uversky Vladimir
Electronic publication date: 2024 Dec 4
Publication date: 2024
Volume: 12
Electronic Location ID: e18591
Received 2024 May 16; Accepted 2024 Nov 5
Copyright: © 2024 Zhu et al.
Copyright year: 2024
Copyright holder: Zhu et al.
License: This is an open access article distributed under the terms of the Creative Commons Attribution License, which permits unrestricted use, distribution, reproduction and adaptation in any medium and for any purpose provided that it is properly attributed. For attribution, the original author(s), title, publication source (PeerJ) and either DOI or URL of the article must be cited.
License URL: https://creativecommons.org/licenses/by/4.0/

Keywords: Acute myocardial infarction, Apoptosis, Biomarker, Machine learning, Immune infiltration

Funding: National Natural Science Foundation of China 82100403 Zhongnan Hospital of Wuhan University Science, Technology and Innovation Seed Fund znpy2019010 This study was funded by a grant from the National Natural Science Foundation of China (No. 82100403) and the Zhongnan Hospital of Wuhan University Science, Technology and Innovation Seed Fund (No. znpy2019010, Huanhuan Cai). The funders had no role in study design, data collection and analysis, decision to publish, or preparation of the manuscript.

==============================
Background

Acute myocardial infarction (AMI) is one of the most serious cardiovascular diseases. Apoptosis is a type of programmed cell death that causes DNA degradation and chromatin condensation. The role of apoptosis in AMI progression remains unclear.

Methods

Three AMI-related microarray datasets (GSE48060, GSE66360 and GSE97320) were obtained from the Gene Expression Omnibus database and combined for further analysis. Differential expression analysis and enrichment analysis were performed on the combined dataset to identify differentially expressed genes (DEGs). Apoptosis-related genes (ARGs) were screened through the intersection of genes associated with apoptosis in previous studies and DEGs. The expression pattern of ARGs was studied on the basis of their raw expression data. Three machine learning algorithms, Least Absolute Shrinkage and Selection Operator (LASSO), support vector machine-recursive feature elimination (SVM-RFE), and Random Forest (RF) were utilized to screen crucial genes in these ARGs. Immune infiltration was estimated by single sample gene set enrichment analysis (ssGSEA). Corresponding online databases were used to predict miRNAs, transcription factors (TFs) and therapeutic agents of crucial genes. A nomogram clinical prediction model of the crucial genes was constructed and evaluated. The Mendelian randomization analysis was employed to investigate whether there is a causal relationship between apoptosis and AMI. Finally, an AMI mouse model was established, and apoptosis in the hearts of AMI mice was assessed via TUNEL staining. qRT-PCR was employed to validate these crucial genes in the hearts of AMI mice. The external dataset GSE59867 was used for further validating the crucial genes.

Results

Fifteen ARGs (GADD45A, DDIT3, FEZ1, PMAIP1, IER3, IFNGR1, CDKN1A, GNA15, IL1B, EREG, BCL10, JUN, EGR3, GADD45B, and CD14) were identified. Six crucial genes (CDKN1A, BCL10, PMAIP1, IL1B, GNA15, and CD14) were screened from ARGs by machine learning. A total of 102 miRNAs, 13 TFs and 23 therapeutic drugs were predicted targeting these crucial genes. The clinical prediction model of the crucial genes has shown good predictive capability. The Mendelian randomization analysis demonstrated that apoptosis is a risk factor for AMI. Lastly, the expression of CDKN1A, CD14 and IL1B was verified in the AMI mouse model and external dataset.

Conclusions

In this study, ARGs were screened by machine learning algorithms, and verified by qRT-PCR in the AMI mouse model. Finally, we demonstrated that CDKN1A, CD14 and IL1B were the crucial genes involved in apoptosis in AMI. These genes may provide new target for the recognition and intervention of apoptosis in AMI.

Introduction

Acute myocardial infarction (AMI) is a prevalent and severe manifestation of coronary artery disease, leading to a substantial number of deaths, with over 2.4 million fatalities in the United States and more than 4 million deaths annually in Europe and northern Asia (Reed, Rossi & Cannon, 2017). About 70% of fatal AMI is caused by coronary artery occlusion caused by atherosclerosis (Mechanic, Gavin & Grossman, 2024). Given that adult cardiomyocytes have limited proliferation capacity (Becker & Hesse, 2020), the extent of cardiomyocytes death serves as a pivotal prognostic indicator subsequent to AMI. Therefore, reducing myocardial cell death has become a critical component to improve the prognosis of AMI. AMI is typically diagnosed by electrocardiogram, coronary angiography, and high-sensitivity cardiac troponin I testing (DeFilippis et al., 2019). However, traditional diagnostic methods still have some limitations in identifying early myocardial infarction (Braunwald, 2012). Thus, discovering novel biomarkers of AMI and investigating the molecular mechanisms underlying the AMI process are of paramount importance.

Apoptosis is considered to be a form of programmed cell death. Its main function is to remove excess or abnormal cells, thereby maintaining the internal environment’s homeostasis (Ketelut-Carneiro & Fitzgerald, 2022). In addition to occurring under physiological conditions, apoptosis can also occur under pathological conditions, such as during viral infections, oxidative stress, ischemia-reperfusion injury, and tumor (Zhang et al., 2024; Winter, Yadav & Rutter, 2022; Kvansakul, 2017).

Apoptosis is known to play a critical role in the progression of various human diseases, and its investigation in the field of cardiovascular disease is steadily advancing. One study demonstrated that liguzinediol, a traditional Chinese medicine, inhibited myocardial apoptosis and improved cardiac function in rats with heart failure (Li et al., 2014). Another study demonstrated that anthocyanins can inhibit apoptosis of cardiac cells induced by cardiac ischemia-reperfusion (Skemiene et al., 2013). Moreover, inhibition of caspase has been shown to prevent cardiac dysfunction caused by cardiac apoptosis in septic rats (Neviere et al., 2001). These findings suggest that apoptosis plays a critical role in cardiovascular diseases. Identifying apoptosis-related genes may offer novel therapeutic targets for the treatment of AMI.

The aim of this study is to identify apoptosis-related genes (ARGs) in AMI through transcriptome dataset analysis, and to explore the biological processes involving ARGs, the factors regulating ARGs expression, their disease predictive capabilities in clinical settings, the causal relationship between AMI and apoptosis, and whether ARGs can serve as regulatory targets and diagnostic markers for AMI. AMI-related datasets (GSE48060, GSE66360, and GSE97320) were merged and analyzed using bioinformatics methods. Based on the original gene expression data, ARGs in AMI were screened, and the characteristic genes among these ARGs were further identified using machine learning algorithm. Then, receiver operating characteristic (ROC) analysis was conducted on the characteristic genes to evaluate their diagnostic value for AMI, and gene set enrichment analysis (GSEA) analysis was performed to identify their involvement in pathways related to AMI. The immune infiltration landscape of the characteristic genes was explored. Next, the miRNAs, transcription factors (TFs) and therapeutic agents that target the characteristic genes were predicted. The Mendelian randomization analysis was employed to investigate whether there is a causal relationship between apoptosis and AMI. Finally, the expression of these characteristic genes was further validated in AMI mouse model and external dataset. This study aims to investigate the molecular mechanism of apoptosis in AMI and identify new biomarkers and therapeutic targets for AMI.

Methods

Data source

The AMI-related datasets (GSE48060, GSE66360, GSE97320 and GSE59867) were obtained from the Gene Expression Omnibus (GEO) database. GSE48060, GSE66360, and GSE97320 are all based on the GPL570 platform ((HG-U133_Plus_2) Affymetrix Human Genome U133 Plus 2.0 Array). GSE59867 is based on the GPL6244 platform ((HuGene-1_0-st) Affymetrix Human Gene 1.0 ST Array (transcript (gene) version)). The samples in GSE48060, GSE66360, GSE97320 and GSE59867 were obtained from the whole blood of AMI patients and healthy individuals. Table 1 shows the information of these datasets. The GWAS data source used for the two-sample Mendelian randomization analysis is from the IEU OpenGWAS project (https://gwas.mrcieu.ac.uk/). Simultaneously, apoptosis-related GWAS data labeled as “prot-a-1373” (3,301 samples) and AMI-related GWAS data labeled as “ukb-e-I21_CSA” (374 case samples and 8,502 control samples) were obtained.

Table 1 Information of the datasets obtained from GEO database.

GEO Accession	Species	Platform	Control group	AMI group	
GSE48060	Homo sapiens	GPL570	21	26	
GSE97320	Homo sapiens	GPL570	3	3	
GSE66360	Homo sapiens	GPL570	50	49	
GSE59867	Homo sapiens	GPL6244	46	111	

Differential expression analysis

GSE48060, GSE66360, and GSE97320 were utilized for screening differentially expressed genes (DEGs). The three datasets were preprocessed, and when the 99th percentile of the data is greater than 100 or the difference between the maximum and minimum values is greater than 50 and the 25th percentile is greater than 0, a log2 transformation was performed on the data. The datasets were then merged, and batch effects in the merged dataset were eliminated using the ComBat function from the R package “sva” (R version 4.2.1). The merged dataset was analyzed for variance using the “limma” R package to identify the DEGs between the AMI and control groups based on the cut-off criteria P < 0.05 & |logFC| > 0.5849625 (FoldChange = 1.5). The volcano plot of DEGs and the heat map based on the top 50 DEGs with P-values were produced by the online bioinformatics analysis platform Sangerbox 3.0 (http://www.sangerbox.com).

Functional and pathway enrichment analysis

The R package “clusterProfiler” was employed for performing The Gene Ontology Consortium (2015) (GO) analysis and Kyoto Encyclopedia of Genes and Genomes (KEGG) pathway enrichment analysis for the identified DEGs. Pathways with a P-value < 0.05 were considered to be statistically significant.

Expression pattern of the differentially expressed ARGs in AMI

A total of 142 genes associated with apoptosis, including drivers, suppressors, and markers, were obtained from previous studies (File S1). Differentially expressed ARGs were obtained by taking the intersection of the DEGs and the genes associated with apoptosis. The trend of gene expression of the differentially expressed ARGs between the AMI and control groups was visualized using a violin plot. The correlation based on Pearson correlation analysis between the differentially expressed ARGs was demonstrated using heat map and PPI network. STRING database was used to construct protein-protein interaction (PPI) network, and Cytoscape software was used to visualize the PPI network. To determine the biological functions of these differentially expressed ARGs, GO annotation and KEGG pathway enrichment analysis were performed.

Screening for crucial ARGs using machine learning algorithms

Three machine learning algorithms, including least absolute shrinkage and selection operator (LASSO) algorithm (Wu et al., 2022), support vector machine-recursive feature elimination (SVM-RFE) algorithm (Sanz et al., 2018), and Random Forest (RF) algorithm (Yifan, Jianfeng & Jun, 2021) were used to screen characteristic genes among the differentially expressed ARGs. LASSO is a type of linear regression that includes an L1 regularization term to shrink the coefficients of less important features (genes) to zero, effectively performing both variable selection and regularization. A LASSO regression model was constructed using the R package “glmnet”. The selection of key genes was achieved through 10-fold cross-validation of the penalty parameters (λ), which controls the strength of regularization. The SVM-RFE algorithm is a machine learning technique designed to enhance feature selection while minimizing overfitting. It works by recursively removing the least important features and re-training the model to optimize performance. The SVM-RFE algorithm, executed by the “e1071” package in R, removed redundant genes and from the dataset and focusing on the features that contribute the most to classification to select key genes. The SVM-RFE model was implemented with feature subset sizes of two, four, six, eight, and increments of three from 10 to 40, using 10-fold cross-validation to optimize feature selection. The RF algorithm is an ensemble learning method that constructs multiple decision trees during training and aggregates their results for classification or regression tasks. It ranks the importance of each gene by evaluating its contribution to improving the classification accuracy across the trees. The RF algorithm was applied with 500 trees to identify important features. The model was optimized by selecting the number of trees with the lowest error rate. Feature importance was ranked using Mean Decrease Gini, and genes with importance scores above two were selected as key genes for further analysis. Finally, the crucial ARGs were obtained by intersecting the key genes identified by the three machine learning algorithms.

Diagnostic value and related biological process of crucial ARGs

To evaluate the diagnostic potential of the identified crucial ARGs, ROC curves were generated using the merged dataset, and the area under the curve (AUC) was calculated to assess their ability to distinguish between AMI and control samples. Furthermore, GSEA was performed using the KEGG database to obtain a deeper understanding of the biological processes linked to the characteristic genes. The samples were stratified into high and low-expression groups based on the expression levels of the characteristic genes.

Identification of the immune infiltration landscape in AMI

Single sample gene set enrichment analysis (ssGSEA) was performed to investigate the immune characteristics in AMI including immune cell infiltration and immune response activity. The 23 signatures of immune cell infiltration and 17 signatures of immune response activity were obtained from “The Immunology Database and Analysis Portal” (https://www.immport.org/home). Additionally, these two immune characteristic categories of the identified characteristic genes were also analyzed.

Prediction of the crucial related miRNAs, transcription factors, and therapeutic drugs

In order to find potential ways to regulate characteristic genes, the miRNAs, transcription factors (TFs) and therapeutic drugs of the characteristic genes were predicted using online databases. The miRNAs of characteristic genes were predicted by online databases miRDB (https://mirdb.org/), TargetScan (https://www.targetscan.org/vert_80/), miRWalk (http://mirwalk.umm.uni-heidelberg.de/) and TarBase (https://dianalab.e-ce.uth.gr/html/diana/web/index.php?r=tarbasev8), among which the miRNAs predicted by three or more databases simultaneously were considered to target these characteristic genes. The online databases TRRUST (https://www.grnpedia.org/trrust/), RegNetwork (Liu et al., 2015), JASPAR (https://jaspar.genereg.net/) and ENCODE (http://genome.ucsc.edu/ENCODE/) were used to predict TFs. Two or more simultaneous predictions are considered to be the TFs target the characteristic genes. The therapeutic drugs targeting the characteristic genes were predicted using the CTD database (http://ctdbase.org/). Finally, interaction network visualization of all the predicted results was performed using Cytoscape software.

Construction and evaluation of the nomogram model

Using the “caret” package in R, the dataset was randomly split into training and testing sets in a 7:3 ratio. Based on the training set, a nomogram clinical prediction model for the crucial genes were constructed using the “rms” package in R. The goodness of fit of the model was assessed using a calibration curve. Additionally, the diagnostic efficiency of the model was assessed using decision curve analysis (DCA) and clinical impact curve in both the testing and training sets.

Two sample Mendelian randomisation analysis

To further investigate the relationship between apoptosis and AMI, two sample mendelian randomization was employed. The GWAS database IEU OpenGWAS project (https://gwas.mrcieu.ac.uk/) was utilized to search for exposure factor data and outcome data. The “TwoSampleMR” R package (Version 0.5.6) was used to perform two-sample Mendelian randomization analysis. In the exposure factor data, SNPs with p < 1e−05 were selected as instrumental variables (IV). Additionally, to ensure the selection of independent SNPs and minimize the effect of linkage disequilibrium (LD) on the results, the LD parameter R2 was set to 0.001, and the genetic distance was set to 10,000 kb. Finally, the strength of the association between the instrumental variables and the exposure factors was assessed using the F-statistic, and SNPs with an F-statistic >10 were included as IVs. The random-effect inverse variance weighted (IVW) was used as the primary method for assessing causal effects. Subsequently, Cochrane’s Q-test was used to assessthe heterogeneity of the results. MR Egger regression was used to assess the pleiotropy of the IVs. The leave-one-out sensitivity analysis was used to evaluate the impact of individual IVs on the overall results. Finally, the results of the Mendelian randomization analysis were visualized through forest plots and scatter plots.

Construction of the AMI mouse model

Animal experiments in this study were conducted following the Guide for the Care and Use of Laboratory Animals (NIH Pub. No. 85-23, revised 1996). The animal experiment in this study was approved by the Experimental Animal Ethics Committee of Zhongnan Hospital of Wuhan University (Ethics approval number: ZN2023201). This study was carried out in compliance with the ARRIVE guidelines. Twelve male C57BL/6 mice (8 weeks old) were purchased from Beijing Vital River Laboratory Animal Technology Co., Ltd. The mice were randomly divided into the sham group and the AMI group. The mice were anesthetized by inhalation of 2% isoflurane. The AMI mouse model was established by ligating the left anterior descending coronary artery. After 24 h of modeling, mice from both the sham group and the AMI group were euthanized, and their hearts were collected for further experiments.

TUNEL staining

Apoptosis in the heart was assessed using TUNEL staining. After obtaining the hearts, they were soaked in 10% formalin, followed by dehydration and paraffin embedding. Subsequently, the hearts were sectioned into 4 μm slices. The paraffin-embedded slices were deparaffinized in xylene and then washed with PBS. The slices were treated with Proteinase K solution at 37 °C for 30 min, followed by PBS washes. The samples were incubated in TUNEL detection solution containing TdT enzyme at 37 °C for 1 h. After being washed with PBS, the samples were incubated in DAPI at 37 °C for 10 min to stain the cell nucleus. The stained samples were observed and captured using an inverted fluorescence microscope (Olympus, Tokyo, Japan). The TUNEL stained cells were quantitatively analyzed by Image J (Version 1.53e) software.

Real-time quantitative polymerase chain reaction

The expression of crucial ARGs in the AMI mouse model was validated through real-time quantitative polymerase chain reaction (qRT-PCR). The total RNA from the heart tissues of the Sham group and AMI group mice was extracted using Trizol (Thermo Fisher, Waltham, MA, USA). Reverse transcription was performed using Hifair III 1st Strand cDNA synthesis SuperMix for qRT-PCR (Ye Sen, China). In qRT-PCR, Hieff UNICON universal blue qPCR SYBR Green Master Mix (Ye Sen, China) was used to fluorescently stain cDNA. The ΔCt values of genes were detected by Bio-Rad CFX96 Real-time PCR system. The reference gene was GAPDH. The relative fold changes of genes were calculated using the 2−ΔΔCt method. File S2 includes the all the primer sequence of qRT-PCR. The data were expressed as the mean values of standard error of the mean (SEM) and were derived from at least three independent experiments.

Further validation of crucial ARGs in external dataset

The GSE59867 dataset contains transcriptomic data from periperal blood mononuclear cells of AMI patients and healthy individuals. The expression of the crucial ARGs in AMI was studied using raw expression data from GSE59867. The independent sample T test was used to compare the AMI group and the control group. P < 0.05 was considered statistically significant.

Results

Identification of DEGs and key pathways in AMI

In the merged dataset (GSE48060, GSE66360, and GSE97320), a total of 657 DEGs were identified based on the cut-off criteria of P < 0.05 and |logFC| > 0.5849625 (FoldChange = 1.5). File S3 includes the detailed information of the DEGs. Figure 1A showed the volcano plot of the DEGs, in which several genes with significant differences were marked. In the merged dataset, DEGs with top50 P-values were shown by heat map (Fig. 1B). KEGG pathway enrichment analysis and GO annotation analysis were utilized to identify the molecular biological functions of the DEGs using the R package “clusterProfiler”. Fifteen KEGG pathways with significant differences were screened and shown by circle plot (Fig. 1C), including “Osteoclast differentiation”, “Leishmaniasis”, “NF-kappa B signaling pathway”, “TNF signaling pathway”, “IL-17 signaling pathway”, “Legionellosis”, “Rheumatoid arthritis”, “Phagosome”, “Amoebiasis”, “Chemokine signaling pathway”, “Toll-like receptor signaling pathway”, “C-type lectin receptor signaling pathway”, “Tuberculosis”, and “Apoptosis”. Bubble plots were used to present the top 15 results of GO annotation (Figs. 1D–1F). Specifically, in the Biological Process (BP) category, DEGs were highly enriched in “immune system process”, “immune response”, and “cell activation”. In the cellular component (CC) category, DEGs were highly enriched in “endomembrane system”, and “vesicle”. In the Molecular Function (MF) category, DEGs were highly enriched in “identical protein binding”, and “signaling receptor binding”.

Figure 1 Identification of DEGs and key pathways in AMI.

(A) Volcano plot of the merged dataset. (B) Heatmap of the top 50 DEGs in the merged dataset. (C) Circle plot of DEGs in KEGG pathway enrichment analysis (TOP15). (D) Bubble plot of DEGs in the term of GO BP (TOP15). (E) Bubble plot of DEGs in the term of GO CC (TOP15). (F) Bubble plot of DEGs in the term of GO MF (TOP15).

Expression pattern of the differentially expressed ARGs in AMI

By intersecting the DEGs with genes associated with apoptosis in previous studies, fifteen differentially expressed ARGs (GADD45A, DDIT3, FEZ1, PMAIP1, IER3, IFNGR1, CDKN1A, GNA15, IL1B, EREG, BCL10, JUN, EGR3, GADD45B, and CD14) were screened out (Fig. 2A). A violin plot was used to display the expression differences of the differentially expressed ARGs between control group and AMI group in the merged dataset (Fig. 2B). Except for FEZ1, the expression of the other 14 ARGs was increased in AMI group. Correlation analysis showed that most of the 15 differentially expressed ARGs were positively correlated, while a few were negatively correlated (GADD45A and FEZ1, EREG and FEZ1, GADD45B and FEZ1) (Fig. 2C). A PPI network of the ARGs, including 12 nodes and 21 edges, was shown in Fig. 2D. GO annotation analysis and KEGG pathway enrichment analysis were used to explore the molecular biological mechanisms involved in these differentially expressed ARGs (Figs. 2E and 2F). The results of GO annotation showed that these genes were mainly enriched in “response to stress”, “regulation of response to stress”, “apoptosis process”, and “programmed cell death”. KEGG pathway enrichment analysis showed that these genes were involved in pathways such as “Colorectal cancer”, “MAPK signaling pathway”, “Apoptosis”, and “p53 signaling pathway”.

Figure 2 Expression pattern of differentially expressed ARGs.

(A) Venn diagram of ARGs. (B) Violin plot of gene expression of differentially expressed ARGs in the merged dataset. (C) Correlation heat map of differentially expressed ARGs in the merged dataset. (D) PPI network of differentially expressed ARGs. (E) Bubble plot of differentially expressed ARGs in GO annotation (TOP15). (F) Circle plot of differentially expressed ARGs in KEGG pathway enrichment analysis (TOP15). *P < 0.05; **P < 0.01; ***P < 0.001. ****P < 0.0001.

Screening for the crucial ARGs using machine learning algorithms

Three machine learning algorithms were used to screen characteristic genes in these differentially expressed ARGs. Nine biomarkers, including GADD45A, FEZ1, PMAIP1, CDKN1A, IL1B, BCL10, JUN and CD14, were screened out from these ARGs by LASSO regression (Fig. 3A). Fifteen biomarkers (GADD45A, DDIT3, FEZ1, PMAIP1, IER3, IFNGR1, CDKN1A, GNA15, IL1B, EREG, BCL10, JUN, EGR3, GADD45B, CD14) were identified by SVM-RFE algorithm (Fig. 3B). Nine biomarkers (GNA15, CDKN1A, DDIT3, PMAIP1, IFNGR1, IL1B, CD14, GADD45B, and BCL10) with importance scores >2 were selected by the RF algorithm (Figs. 3C and 3D). Finally, the intersection of the three machine learning results yielded six characteristic genes (CDKN1A, BCL10, PMAIP1, IL1B, GNA15, and CD14). The Venn diagram of the intersection result was shown in Fig. 3E.

Figure 3 Screening for characteristic genes using machine learning algorithms.

(A) Plot of λ selection in the LASSO model. (B) Plot of SVM-RFE machine learning result. (C and D) Plot of RF algorithm and ARGs ranked by importance score. (E) Identification of the characteristic genes.

Diagnostic value and related biological process of the crucial ARGs

ROC analysis was performed on six characteristic genes, and their AUC values were calculated (Figs. 4A–4F). In the ROC curve, genes with AUC > 0.7 and P < 0.05 were considered to have diagnostic value. Among these, five genes, including IL1B (AUC = 0.7968, P < 0.0001), GNA15 (AUC = 0.7737, P < 0.0001), CDKN1A (AUC = 0.7416, P < 0.0001), BCL10 (AUC = 0.7355, P < 0.0001), and CD14 (AUC = 0.7350, P < 0.0001), were identified as having diagnostic value for AMI. To gain further insight into the pathways associated with these characteristic genes, GSEA analysis were performed. Figures 4G–4L showed the GSEA results based on the KEGG pathway for the six characteristic genes. According to enrichment score (ES) and P value, the pathways with top5 enrichment level in each gene were selected for display.

Figure 4 ROC curves and GSEA analysis of the characteristic genes.

(A–F) ROC curves of the characteristics genes. (G–L) Related biological process of the characteristic genes.

Identification of the immune infiltration landscape in AMI

The analysis of immune infiltration showed that the degree of immune cell infiltration and immune response activity was increased in AMI. In the AMI group, there was an increased infiltration of sixteen immune cells (activated CD4 T cell, activated dendritic cell, CD56bright natural killer cell, eosinophil, Gamma delta T cell, immature dendritic cell, MDSC, macrophage, mast cell, monocyte, natural killer T cell, natural killer cell, neutrophil, plasmacytoid dendritic cell, T follicular helper cell, and Type 1 T helper cell), whereas one immune cell (Activated CD8 T cell) showed decreased infiltration (Fig. 5A). Additionally, a total of thirteen immune responses were up-regulated in AMI group, including cytokines, chemokines, cytokine receptors, TGFb family members, chemokine receptors, BCRSignaling pathway, TNF Family members receptors, NaturalKiller cell cytotoxicity, TNF family members, interleukins receptors, interferon receptor, antimicrobials, and interleukins (Fig. 5B). Subsequently, the correlation between characteristic genes and immune infiltration was investigated. It was found that mast cells and neutrophil exhibited the strongest positive correlation with IL1B (0.73), while activated CD8 T cells showed the strongest negative correlation (−0.24) (Fig. 5C). Furthermore, chemokine receptor displayed the strongest positive correlation with IL1B (0.63), and TGFb family member receptor exhibited the strongest negative correlation with CDKN1A (−0.25) (Fig. 5D).

Figure 5 Identification of the immune infiltration landscape in AMI.

(A) Box plot of immune cell infiltration in AMI and control. (B) Box plot of immune response activity in AMI and control. (C) Heat map of immune cell infiltration in the six characteristic genes. (D) Heat map of immune response activity in the six characteristic genes. *P < 0.05; **P < 0.01; ***P < 0.001.

Prediction of the crucial ARGs related miRNAs, transcription factors, and targeted therapeutic drugs

The results showed that a total of 102 miRNAs targeted the six characteristic genes, among which hsa-miR-298 targeted both GNA15 and CDKN1A (Fig. 6A). In addition to GNA15, whose TFs was not predicted, a total of thirteen TFs were predicted to regulate the five characteristic genes (IL1B, CDKN1A, CD14, PMAIP1, and BCL10), among which CEBPA can target both CDKN1A and CD14 (Fig. 6B). A total of 23 therapeutic drugs, which can simultaneously target at least five or more characteristic genes, have been screened out, including genistein, calcitriol, amphetamine, phenobarbital, etc (Fig. 6C).

Figure 6 Prediction of ARG-related miRNAs, transcription factors, and therapeutic drugs.

(A) The miRNA-mRNA network. (B) The TF-mRNA network. (C) The drug-mRNA network. (A–C) Orange diamond patterns represent the characteristics genes; blue ellipse patterns represent miRNAs; purple round rectangle patterns represent TFs; yellow triangle patterns represent the therapeutic drugs.

Construction and evaluation of the nomogram model

Six crucial genes CDKN1A, BCL10, PMAIP1, IL1B, GNA15, and CD14 screened by machine learning were considered the risk factor variables for AMI. A nomogram clinical prediction model was constructed based on the raw expression data of these genes in the dataset (Fig. 7A). The calibration curve indicated that the model has good fitting performance (Fig. 7B). In addition, the DCA results (Figs. 7C and 7D) and clinical impact curves (Figs. 7E and 7F) in both the training and testing sets showed that the model has good diagnostic efficiency.

Figure 7 Construction and evaluation of the nomogram model.

(A) Construction of a nomogram to predict AMI risk based on the six crucial genes. (B) Calibration curves of the nomogram. (C) DCA results of the nomogram in the training set. (D) DCA results of the nomogram in the testing set. (E) Clinical impact curves of the nomogram in the training set. (F) Clinical impact curves of the nomogram in the testing set.

Two sample Mendelian randomization analysis

Exposure factor data related to apoptosis, labeled as “prot-a-1373”, and outcome data related to AMI, labeled as “ukb-e-I21_CSA”, were obtained from the IEU OpenGWAS project. After screening, a total of 17 IVs were included in the exposure factor data. The IVW result of the two-sample Mendelian randomization analysis indicate a causal relationship between apoptosis and AMI (P < 0.05), with apoptosis being a risk factor for AMI (OR = 1.426; 95% CI [1.042–1.953]) (Fig. 8A). The results of leave-one-out sensitivity analysis confirmed the absence of outlier variable among the IVs (Fig. 8B). The Cochrane’s Q-test results showed P > 0.05, indicating no heterogeneity among the IVs (Fig. 8C). Pleiotropy analysis demonstrates that there is no pleiotropy among the IVs. Ultimately, different algorithms have all demonstrated that apoptosis can increase the risk of acute myocardial infarction (Fig. 8D). The complete Mendelian randomization analysis results can be found in File S3.

Figure 8 Two sample Mendelian randomization analysis.

(A) Forest plot for Mendelian randomization analysis of apoptosis and AMI. (B) Forest plot of leave-one-out sensitivity analysis. (C) Funnel plot of heterogeneity analysis. (D) Scatter plot for Mendelian randomization analysis of apoptosis and AMI.

Apoptosis detection and validation of the crucial ARGs in AMI mouse model

A mouse AMI model was successfully established after 24 h ligation of the left anterior descending coronary artery (Fig. 9A). TUNEL staining was used to assess apoptosis in the hearts of both the AMI and Sham groups of mice. The results indicated a significant increase in apoptosis in the hearts of the AMI group compared to those in the Sham group (Fig. 9B). Six crucial ARGs (CDKN1A, BCL10, PMAIP1, IL1B, GNA15, and CD14) were validated in AMI mouse model (Fig. 9C). The qRT-PCR results of the AMI mouse model revealed that four crucial ARGs (CDKN1A, IL1B, CD14, and PMAIP1) were up-regulated in the AMI group. Furthermore, their expression trends were consistent with the results obtained from the bioinformatics analysis. Finally, these four genes were considered to be involved in the progression of apoptosis in AMI and to be an apoptotic biomarker in the blood of AMI patients.

Figure 9 Apoptosis detection and validation of the crucial ARGs in AMI mouse model.

(A) Photographs of the hearts of AMI mice and Sham mice. (B) TUNEL staining of heart slices from AMI mice and Sham mice. (C) Validation of the ARGs in AMI mouse model. *P < 0.05; **P < 0.01; ****P < 0.0001.

Further validation of crucial ARGs in external dataset

The six differentially crucial ARGs (CDKN1A, BCL10, PMAIP1, IL1B, GNA15, and CD14) were further evaluated through the use of external dataset. In GSE59867, CDKN1A, IL1B, GNA15 and CD14 were highly expressed in the AMI group and there was no significant difference on BCL10 and PMAIP1 between AMI group and control group (Fig. 10).

Figure 10 Further validation of crucial ARGs in external dataset.

(A–F) Validation of the crucial ARGs in GSE59867. ***P < 0.001; ****P < 0.0001.

Discussion

In this study, we discovered a causal relationship between apoptosis and AMI through Mendelian randomization analysis, and apoptosis was found to exacerbate the development of AMI. Furthermore, we screened 657 DEGs in a merged dataset composed of three AMI-related datasets (GSE48060, GSE66360, and GSE97320). The KEGG pathway enrichment analysis of these DEGs indicated that apoptosis is one of the major pathways in AMI. Next, we further identified 15 ARGs (GADD45A, DDIT3, FEZ1, PMAIP1, IER3, IFNGR1, CDKN1A, GNA15, IL1B, EREG, BCL10, JUN, EGR3, GADD45B, and CD14) among these DEGs. These ARGs are primarily enriched in the MAPK signaling pathway, p53 signaling pathway, and NF-κB signaling pathway. Through machine learning, we further identified six crucial genes (CDKN1A, BCL10, PMAIP1, IL1B, GNA15, and CD14) among the ARGs. Through ROC curves, we demonstrated that these six crucial ARGs have good diagnostic performance in AMI. Subsequently, we established a mouse AMI model and found a significant increase in apoptosis in the hearts of AMI mice through TUNEL staining. We validated these six crucial ARGs in the hearts of AMI mice through qRT-PCR and in the AMI-realted external dataset, ultimately confirming that the expression of three of these genes (CDKN1A, IL1B and CD14) was up-regulated.

AMI is an acute coronary syndrome caused by coronary artery obstruction leading to myocardial cell necrosis. With optimization of the diagnosis and treatment of AMI, the in-hospital mortality rate of ST-segment elevation myocardial infarction (STEMI) has now been reduced to 6–7% (Boateng & Sanborn, 2013). The main diagnostic methods for AMI are electrocardiogram and the measurement of myocardial necrosis serum biomarkers. In AMI cases, 30% are ST-segment elevation myocardial infarction (STEMI), while 70% are non-ST-segment elevation myocardial infarction (NSTEMI) with varying electrocardiogram characteristics (Bhatt, Lopes & Harrington, 2022). Meanwhile, the cardiac injury-specific biomarker cTn can also increase in other diseases, such as chronic kidney disease (Park et al., 2017). In terms of the treatment of AMI, early reperfusion therapy can greatly avoid myocardial cell death and limit the expansion of infarction area (Zhang & Huo, 2011). But, reperfusion also can induce inflammatory reactions and thus expand the AMI-related damage (Sanchez-Hernandez et al., 2020). Therefore, in-depth research on the pathogenesis of AMI and the search for new gene targets for diagnosis and treatment are of great significance. Apoptosis, as a programmed cell death mode closely related to myocardial infarction, has received wide extensive attention. We used bioinformatics methods to mine and analyze AMI-related datasets, aiming to explore the molecular mechanisms of ARGs in AMI and identify new apoptotic targets.

In our study, 657 DEGs were selected from a merged dataset of AMI-related datasets, including GSE48060, GSE66360, and GSE97320. These genes are mainly enriched in pathways such as the “NF-κB signaling pathway”, “TNF signaling pathway”, “IL-17 signaling pathway”, “apoptosis”, “immune response”, “endomembrane system”, “identical protein binding”, and “signaling receptor binding”. NF-κB, TNF, and IL-17 signaling pathways are classical inflammation-related pathways, indicating that the progression of AMI is closely related to inflammation response. The immune infiltration analysis in our study also indicates a significant increase in the immune response activity of the TNF family in AMI. In addition, studies have suggested that these pathways are associated with apoptosis. A study has shown that Corin can inhibit H2O2-induced apoptosis in cardiomyocytes by activating the NF-κB pathway (Li et al., 2018). In addition, TNF-α has been shown to be closely associated with apoptosis in various cardiovascular diseases such as dilated cardiomyopathy and septic cardiomyopathy (Zheng et al., 2020; Das et al., 2010). IL-17, an inflammatory cytokine secreted by immune cells, can act on different targets to regulate the inflammatory response (Mora-Ruiz et al., 2019). It is worth noting that IL-17 can induce apoptosis by activating iNOS in cardiomyocytes (Su et al., 2016).

Apoptosis, a programmed cell death mechanism, was proposed as early as 1972 and since then, the genes involved in the apoptotic pathway have been extensively studied. There are various pathways to induce apoptosis, such as the intrinsic pathway, extrinsic pathway, and endoplasmic reticulum stress pathway. Similarly, in different diseases, the pathways that induce apoptosis seem to be different (Pinkaew et al., 2017; Pitti et al., 1996; Schuler & Green, 2001). Therefore, we used bioinformatics to identify the genes and pathways that regulate apoptosis in AMI. In this study, we employed machine learning methods to screen six crucial ARGs (BCL10, CDKN1A, CD14, GNA15, IL1B, and PAMIP1). We validated these genes using the AMI mouse model and external dataset. Results showed that three crucial ARGs (CDKN1A, IL1B and CD14) were highly expressed in AMI mouse, and the ROC curve indicated that these three genes had good diagnostic value.

In previous studies, apoptosis was often considered to be the result of AMI. However, few studies have discussed whether enhanced apoptosis exacerbates the progression of AMI. In this study, apoptosis was taken as the exposure factor and AMI as the outcome factor, and the causal relationship between them was explored through Mendelian randomization analysis. The results indicated that there was a causal relationship between the activation of apoptosis and the progression of AMI. A study also showed a conclusion that activation of exogenous caspase-8 in mice led to a large number of cardiomyocyte apoptosis, and caused the death of heart failure in mice (Wencker et al., 2003).

miRNAs are a class of non-coding RNAs that play crucial regulatory roles within cells. One of the most common functions of miRNAs is to bind to mRNA, thereby suppressing or degrading it, ultimately influencing gene expression. Moreover, miRNAs have been demonstrated to serve as specific and sensitive biomarkers for various diseases. A research has shown that miR-298 can regulate apoptosis in myocardial infarction by targeting BAX (Zhang, Yu & Yu, 2018). Our study also revealed that hsa-miR-298 can target CDKN1A and GNA15. Hence, we consider miR-298 as a crucial regulatory factor for apoptosis in AMI. In addition, miRNAs that we predicted, such as let-7b-5p (Venugopal et al., 2022), miR-106a-5p (Bye et al., 2016), miR-17-5p (Xue et al., 2019), have also been demonstrated to serve as circulating diagnostic markers for AMI.

CDKN1A encodes a cyclin-dependent kinase inhibitor that can control the cell cycle progression by inhibiting the activity of cyclin-dependent kinase 2, and its expression is strictly regulated by p53 (Engeland, 2022). The pro-apoptotic or anti-apoptotic effects of CDKN1A depend on the specific cell type and cellular environment (Liu, Bishop & Liu, 2003). A study has shown that CDKN1A can inhibit apoptosis by binding to procaspase-3 (Suzuki et al., 2013). There are also a study suggesting that overexpression of CDKN1A can induce apoptosis in ovarian cancer cells (Wu et al., 2002). Additionally, a study suggests that blocking the expression of CD14 in polymorphonuclear neutrophil granulocytes can attenuate the anti-apoptotic effect of lipoteichoic acid (Lotz et al., 2004). IL1B is primarily secreted by macrophages, processed into its active form by CASP1. It can be involved in apoptosis, pyroptosis, inflammation, and more. Researches have reported that IL1B can induce apoptosis in various diseases (Paira et al., 2022; Shao et al., 2020). A study reported that miR-132 can target IL1B in myocardial infarction to inhibit apoptosis of myocardial cells (Zhao et al., 2020), further demonstrating the potential of IL1B as a target for apoptosis intervention and a biomarker. PMAIP1 (NOXA) is a pro-apoptotic member of the BCL-2 protein family, which promotes apoptosis by facilitating the activation of caspase. Current research indicates that PMAIP1 can promote apoptosis in various diseases (Zhao, Liu & Su, 2014; Asuzu et al., 2022; Su et al., 2017). In the upstream of PMAIP1, TP53 has been shown to enhance the apoptotic activity of PMAIP1 (Liang et al., 2019; Nakajima & Tanaka, 2007). Currently, PMAIP1 has become a biomarker of apoptosis. The increase in PMAIP1 further demonstrates the role of apoptosis in the progression of acute myocardial infarction.

This study has restrictions since it only validated the expression of these ARGs in AMI mouse model, and did not further investigate the specific role of these ARGs in AMI apoptosis. However, these ARGs have been shown in previous studies to be involved in apoptosis in different diseases. Our study utilized multiple datasets from the GEO database for analysis and validation, and the use of public databases enhanced the reproducibility of the research findings. However, public datasets also present potential biases. Variations in sample sources, experimental conditions, and data processing across different studies may impact result consistency, leading to the possibility that more key genes may not be identified. Another limitation of this study is the lack of functional validation of the identified genes in human samples. Although bioinformatics analyses provided valuable insights into potential key genes and pathways, experimental validation in human tissues or cells is essential to confirm their roles in disease mechanisms. In the future, it is necessary to collect blood and heart tissues from AMI patients to verify the diagnostic efficacy of these ARGs. Moreover, the GWAS data used in this study are cross-sectional in nature, which limits the causal inference in Mendelian randomization analysis. Therefore, future studies should incorporate longitudinal data to more accurately assess the causal relationship between genes and diseases. Furthermore, the specific mechanisms for these ARGs were not explored in this work. Further studies would be performed in genetically modified animal models in future.

Conclusion

In this study, we identified CDKN1A, CD14 and IL1B as characteristic genes involved in apoptosis in AMI through bioinformatics analysis and validation using the AMI mouse model and the external dataset. This may provide a new basis for the recognition and interference of apoptosis in AMI.

Supplemental Information

Supplemental Information 1 Genes associated with apoptosis from previous studies.

Supplemental Information 2 Primer sequences used in the study.

Supplemental Information 3 The variance within the merged data using PCA analysis.

Supplemental Information 4 ARRIVE 2.0 Checklist.

Supplemental Information 5 The information of differentially expressed genes.

Supplemental Information 6 MIQE checklist.

Supplemental Information 7 Raw data.

We sincerely appreciate the researchers for providing their GEO database information online, and we are truly honored to acknowledge their contributions. We are also grateful to the Sanger Box online biomedical data analysis tool for simplifying the analysis process (http://sangerbox.com).

Online database addresses

https://www.ncbi.nlm.nih.gov/.

https://www.ncbi.nlm.nih.gov/geo.

http://www.sangerbox.com/tool.html.

http://bioinformatics.psb.ugent.be/webtools/Venn/.

https://www.gsea-msigdb.org/gsea/msigdb.

https://www.gsea-msigdb.org/gsea.

https://www.immport.org/home.

https://mirdb.org/.

https://www.targetscan.org/vert_80/.

http://mirwalk.umm.uni-heidelberg.de/.

https://dianalab.e-ce.uth.gr/html/diana/web/index.php?r=tarbasev8.

https://www.grnpedia.org/trrust/.

Liu et al. (2015).

https://jaspar.genereg.net/.

http://genome.ucsc.edu/ENCODE/.

http://ctdbase.org/.

https://gwas.mrcieu.ac.uk/.

Abbreviations list

AMI Acute Myocardial Infarction

ARG Apoptosis-Related genes

BP Biological Process

CC Cellular Component

DEG Differentially expressed genes

GEO Gene Expression Omnibus

GO Gene Ontology

GSEA Gene Set Enrichment Analysis

KEGG Kyoto Encyclopedia of Genes and Genomes

LASSO Least absolute shrinkage and selection operator

MF Molecular Function

RF Random forest

ssGSEA Single sample gene set enrichment analysis

SVM-RFE Support vector machine-recursive feature elimination

TF Transcription Factors

Additional Information and Declarations

Competing Interests

Author Contributions

Animal Ethics

Data Availability

The authors declare that they have no competing interests.

Haoyan Zhu conceived and designed the experiments, performed the experiments, analyzed the data, prepared figures and/or tables, and approved the final draft.

Mengyao Li conceived and designed the experiments, prepared figures and/or tables, and approved the final draft.

Jiahe Wu conceived and designed the experiments, analyzed the data, prepared figures and/or tables, and approved the final draft.

Liqiu Yan analyzed the data, prepared figures and/or tables, and approved the final draft.

Wei Xiong performed the experiments, prepared figures and/or tables, and approved the final draft.

Xiaorong Hu analyzed the data, authored or reviewed drafts of the article, and approved the final draft.

Zhibing Lu analyzed the data, authored or reviewed drafts of the article, and approved the final draft.

Chenze Li conceived and designed the experiments, authored or reviewed drafts of the article, and approved the final draft.

Huanhuan Cai conceived and designed the experiments, authored or reviewed drafts of the article, and approved the final draft.

The following information was supplied relating to ethical approvals (i.e., approving body and any reference numbers):

The animal experiment in this study was approved by the Experimental Animal Ethics Committee of Zhongnan Hospital of Wuhan University (Ethics approval number: ZN2023201).

The following information was supplied regarding data availability:

The qRT-PCR results and primer sequences are available in the Supplemental Files.

The raw data used in this study are available at GEO: GSE48060, GSE66360, GSE97320 and GSE59867.

The processed data and analysis scripts are available at GitHub:

- https://github.com/Sou1Beg9ar/AMI-apoptosis/blob/main/Rscript.R.

- Sou1Beg9ar. (2024). Sou1Beg9ar/AMI-apoptosis: R script of apoptosis in AMI (AMI-apoptosis). Zenodo. https://doi.org/10.5281/zenodo.14092716.

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
