# Peer review of "Identification and validation of apoptosis-related genes in acute myocardial infarction based on integrated bioinformatics methods"

_PeerJ, doi:10.7717/peerj.18591_

## Round 0.1 · original submission · Major Revisions

Please address concerns of both reviewers and amend manuscript accordingly

·

Basic reporting

Improvement needed:
1. Update with recent references to provide current context. Include newer studies on apoptosis in pathological conditions from the past five years. For instance, line 91, the statistical facts about AMI related death is a little bit old. Considering citing a more recent article. StatPearls 2023, can be accessed from NCBI (https://www.ncbi.nlm.nih.gov/books/NBK459269/), line 104: references are from over two decades, considering more recent studies.
2. Considering increase the figure quality. For example, figure 3C, and figure 4G -L.

Experimental design

Improvement needed:
1. Need to convey the research question and hypothesis more clearly. For instance, explicitly state the research question and hypotheses at the beginning of the study.
2. Need to add more information to enhance the reproducibility of the study. For instance, provide the source coding scripts: "The raw data used in this study are available in the GEO repository with the primary accession codes GSE48060, GSE66360, and GSE97320. Additionally, the processed data and analysis scripts can be accessed at [provide URL or repository details]."
3. Need to ensure statistical methods and controls are described in detail to validate robustness. For instance, line 147: Need to provide justification for the cut off criteria for the Foldchange. Has the cutoff value been used by previous studies?

Validity of the findings

Improvement needed:
1. Need to provide justification of using the TUNEL staining. Why other staining methods, such as Annexin V Staining (which can differentiate between early and late apoptosis, and necrosis.), Caspase Activity Assays (which can directly measure apoptotic enzyme activity), do/do not fit the purpose of the current research best?
2. Need to clearly describe the machine learning algorithms used (LASSO, SVM-RFE, RF), their implementation and justify the selection criteria for key genes identified by machine learning.

Reviewer 2 ·

Basic reporting

The authors' efforts in this manuscript are commendable. However, please ensure the figure labels are correct and readable. They were challenging to read, which could affect the readers' understanding of the findings.

Experimental design

The manuscript by Haoyan Zhu, Mengyao Li et al. is a testament to their deep understanding and expertise in informatics and cardiovascular biology. Their work on apoptosis and AMI, which have been familiar topics in the field since the late 90s, is highly recognized and appreciated. While apoptosis may provide a new target for cardioprotection during evolving AMI in humans, this is already well known. Therefore, the novelty of the study needs further justification.

Validity of the findings

It would be fascinating to delve deeper into the authors' research methods. For instance, what is the variance within the merged data using statistical methods like PCA analysis of AMI peripheral blood samples before merging? From a bioinformatic point of view, although the consistency of sample selection from the same array platform is appreciated, having only three studies for comparison is insufficient. It is highly recommended that more robust RNA-seq or SC-RNA seq be used to validate the findings, thus opening new avenues for research.

Additional comments

Is there any rationale for the authors' choice of a 1.5-fold change as a cut-off when aiming for a related target or prognosis marker? The heatmap between the control and AMI does not show substantial differences; a change of < 50% may indicate slight differences. Could this be why KEGG Pathway analysis showed many unrelated enrichments, such as Leishmaniasis, Amoebiasis, and Tuberculosis? These factors could affect gene prediction efficiency using ML algorithms. Please provide appropriate justification.
The KEGG pathway from the DEGs data showed apoptosis as one but not the most significant.
From the GO terms, the biological processes are mostly immune responses, which is expected as they involve peripheral blood and inflammation. However, more neutrophil-related terms are intriguing and could be due to the shorter life span of neutrophils and their prognostic value.
If ARG genes overlap with AMI, gene enrichment in GO terms is expected to be seen in terms related to apoptosis or programmed cell death. Is there any additional significance to this analysis? If not, it might be better included in supplementary data.
If these genes are pre-selected using DEG overlaps and are mostly positively correlated, lasso regression (like SVM-Recursive feature elimination) can be compromised when screening biomarkers. How are the authors accounting for this limitation?
MiRNAs are involved in suppressing or degrading the target mRNA, so how do authors explain so many miRNAs associated with selected ARGs? Expression seems to be higher when quantified by RT qPCR. What is the author's response to this?

---

## Round 0.2 · Minor Revisions

Please address remaining issues pointed by the reviewer and amend manuscript accordingly.

·

Basic reporting

N/A

Experimental design

While the methods are detailed, some areas require additional information for reproducibility:
• Provide more details on how the datasets were merged and batch effects corrected (e.g., parameters used in the "sva" package).
• How hyperparameters were selected for the machine learning algorithms.
• Include details on the selection criteria for instrumental variables, statistical tests used, and software packages with versions for the mendelian randomization analysis.

Validity of the findings

Ensure consistency in terminology and acronyms :
• Both "AMI" and "MI" are used interchangeably in lines 369, 375, and 467. For clarity and to avoid confusion, please ensure consistent use of terminology throughout the text.
• In lines 426 and 428, you mention identifying four genes; however, later in line 454, only three genes are included. Additionally, in lines 497 and 499, you refer to four genes again but only specify three, with PMAIP1 being omitted. Please review these sections carefully to ensure consistency in the number of genes discussed and clarify which genes are being referred to in each instance.
Expand the limitations section to discuss:
• The use of public datasets and potential biases.
• The lack of functional validation of the identified genes in human samples.
• The cross-sectional nature of the data limiting causal inferences.

Reviewer 2 ·

Basic reporting

Satisfactory

Experimental design

Satisfactory

Validity of the findings

Satisfactory

---

## Round 0.3 · accepted · Accept

All remaining concerns were addressed, and revised manuscript is acceptable now.